# A High-Performance 2.5 μm Charge Domain Global Shutter Pixel and Near Infrared Enhancement with Light Pipe Technology [note 1]

**DOI:** 10.3390/s20010307

**Published:** 2020-01-06

**Authors:** Ikuo Mizuno, Masafumi Tsutsui, Toshifumi Yokoyama, Tatsuya Hirata, Yoshiaki Nishi, Dmitry Veinger, Adi Birman, Assaf Lahav

**Affiliations:** 1TowerJazz Panasonic Semiconductor Co., Ltd., 800 Higashiyama, Uozu City, Toyama 937-8585, Japan; tsutsui.masafumi@tpsemico.com (M.T.); yokoyama.toshifumi@tpsemico.com (T.Y.); hirata.tatsuya@tpsemico.com (T.H.); nishi.yoshiaki@tpsemico.com (Y.N.); 2Tower Semiconductors, Migdal Haemeq 23105, Israel; Dmitry.Veinger@towerjazz.com (D.V.); adibi@towersemi.com (A.B.); asafla@towersemi.com (A.L.)

**Keywords:** CMOS image sensor, global shutter, charge domain, parasitic light sensitivity, dark current, near infrared, modulation transfer function

## Abstract

We developed a new 2.5 μm global shutter (GS) pixel using a 65 nm process with an advanced light pipe (LP) structure. This is the world’s smallest charge domain GS pixel reported so far. This new developed pixel platform is a key enabler for ultra-high resolution sensors, industrial cameras with wide aperture lenses, and low form factors optical modules for mobile applications. The 2.5 μm GS pixel showed excellent optical performances: 68% quantum efficiency (QE) at 530 nm, ±12.5 degrees angular response (AR), and quite low parasitic light sensitivity (PLS)—10,400 1/PLS with the F#2.8 lens. In addition, we achieved an extremely low memory node (MN) dark current 13 e^−^/s at 60 °C by fully pinned MN. Furthermore, we studied how the LP technology contributes to the improvement of the modulation transfer function (MTF) in near infrared (NIR) enhanced GS pixel. The 2.8 μm GS pixel using a p-substrate showed 109 lp/mm MTF@50% at 940 nm, which is 1.6 times better than that without an LP. The MTF can be more enhanced by the combination of the LP and the deep photodiode (PD) electrically isolated from the substrate. We demonstrated the advantage of using LP technology and our advanced stacked deep photodiode (SDP) technology together. This unique combination showed an improvement of more than 100% in NIR QE while maintaining an MTF that is close to the theoretical Nyquist limit (MTF @50% = 156 lp/mm).

## 1. Introduction

In recent years, there is a strong market demand for small pitch and high performance global shutter (GS) sensors, which can take images without distortion for fast-moving objects. This function is highly desirable for machine vision. Moreover, small GS pixel with high near infrared (NIR) sensitivity is attractive, especially for facial and motion recognition of humans for mobile and automotive use.

The GS CMOS image sensor can be divided into two types. One is the charge domain type, in which charges generated in the PD are stored in MN [1,2]. The other is the voltage domain type, in which charges generated in the PD are amplified in each pixel and then stored in a capacitor as voltage [3,4]. The voltage domain type requires at least two switching transistors and capacitors when correlation double sampling (CDS) is enabled, even though the kT/C noise of the switching transistor remains, whereas the charge domain type realizes true CDS with only one additional memory node (MN). In other words, the charge domain type is more promising for pixel scalability and low noise than the voltage domain type. Therefore, we have focused on the charge domain type.

In the practical use of the GS sensor, it is important to achieve a high signal-to-noise ratio. Since the charge domain GS pixel has additional components such as MN, compared with the rolling shutter pixel; suppressing the dark current generated in the MN is one of the key factors of charge domain GS pixel development. In addition, suppressing PLS is very important even in oblique incident light. For machine vision, large optical format sensors are usually used for high resolution. For mobile use, module height must be low; therefore, small F# lenses are necessary. In addition, when NIR sensitivity is enhanced, cross talk to adjacent pixels, which is measured as the modulated transfer function (MTF), must be discussed since the absorption of NIR light requires deeper silicon than visible light. Particularly in GS pixels, a metal shield is used for suppressing the parasitic light sensitivity (PLS) of MN [5,6]. Its small metal window defined by the tungsten (W) shield leads to diffraction, which makes it difficult for the pixels to have good MTF. Thus, it is desirable that the light will go straight and will enter directly into the silicon without diffraction, despite the small metal window.

In order to meet such requirements, we have developed a small pitch GS pixel. In the past, we already used a 110 nm process to design and fabricate a low noise and high QE 2.8 μm GS pixel [7,8,9]. In this paper, we report a 2.5 μm GS pixel that features extremely low dark current MN and advanced LP structure using a 65 nm process [5]. The same technology was used to study how the LP technology contributes to the improvement of MTF in NIR-enhanced GS pixels using a 2.8 μm pixel.

Section 2 describes the device structure. Section 3 reports the development of the 2.5 μm charge domain GS pixel, whose key factors are a narrow-shaped MN design and LP design. Finally, Section 4 discusses the study of NIR enhancement with LP technology using 2.8 μm and 3.2 μm GS pixels. Conclusions are presented in Section 5.

## 2. Device Structure

Figure 1 shows the circuit schematic of the developed pixels. In order to maximize the active area, there is no row select transistor by using the floating diffusion drive method [10]. Floating diffusion (FD) is shared by two pixels diagonally. TX1 is served as a control line for both a transfer gate from the PD to MN, and a storage gate over MN, in order to reduce a control line for the storage gate. The control line of TX1 is shared vertically. TX1 maintains a negative bias except during charge transfer from PD to MN. TX2 also maintains a negative bias except during charge transfer from MN to FD. This negative bias accumulates holes at the surface of MN, which occupy recombination centers and prevent carrier generation [11].

Figure 2 shows the cross-section schematic of the developed pixel. The MN is covered with the W-shield to protect from incident light [12]. The ultra-thin backend process with three Cu layers using a 65 nm process reduced the optical height from the Si surface to the microlens and realized the light pipe (LP) structure using a high refractive index material [13,14], which contributed to the immunity against oblique light. The core material for the LP is SiN. Since the refractive index of SiN is high, incident light is bent at the SiN surface. In addition, incident light into the LP is totally reflected inside the LP by utilizing the difference in refractive index between SiN and the oxide film, and it is confined in the LP, which greatly improved the AR performance.

The type of Si substrate can be chosen according to the requirements of NIR performance. Figure 3 shows the simple cross-section schematic for (a) an N-type substrate and (b) a P-type substrate. The N-type substrate is superior to the P-type substrate in terms of dark current performance and MTF, because the PD is better electrically isolated from the adjacent PDs and substrate. The isolation between PD and substrate limits NIR QE with the N-type substrate. On the other hand, NIR QE with the P-type substrate depends on its wafer specifications: the concentration of substrate and epitaxial thickness, because PD is not electrically isolated from substrate. When a P-type concentration of substrate is low, the lifetime of electrons in the substrate will increase, which leads to a much higher NIR QE than that with a high P-type concentration of substrate instead of a higher dark current. Therefore, the p-type substrate is promising in terms of NIR enhancement.

## 3. Development of 2.5 μm Charge Domain Global Shutter Pixel

### 3.1. Design Concept for Charge Domain

The placement of PD and MN should be determined by taking the tradeoffs of each performance into account. Table 1 shows the figure of merit for PD and MN as the GS pixel’s charge domain. The light sensitivity of the PD and its angular response (AR) needs to be larger, while that of the MN and its AR needs to be smaller. The full well capacity (FWC) of the MN should be larger than that of the PD in order to store the electrons transferred from the PD without loss.

Our deliberate study led us to choose a narrower MN and a wider PD, namely maximizing the fill factor, which helps to maximize the QE with good AR. Therefore, the challenges of this work are as follows: (1) narrow full pinned MN design to achieve a large FWC and low dark current with good charge transfer from MN to FD, and an (2) LP and metal shield design to keep the sensitivity of MN low even with oblique light.

### 3.2. Memory Node Design

In order to suppress the dark current, we adopted the full pinned MN structure [8,9]. Figure 4 shows the cross-section schematics of (a) conventional MN and (b) full pinned MN. The adopted MN has p-type implants, both under the poly-Si gates, and in the gap between TX1 and TX2, in order to accumulate holes at the surface with negative gate biasing. The surface p-type implant makes the MN’s threshold voltage higher and increases hole accumulation with the negative gate biasing. Due to the lower effectiveness of negative gate biasing in the region in the gap, a high concentration p-type implant to the gap is required in order to accumulate holes in the gap. However, the higher p-type concentration in the gap can make a potential barrier in the gap, which can make it difficult for electrons to transfer from MN to FD. Therefore, the dosage of the p-type implant to the gap should be determined considering FWC and the image lag of the MN.

In order to obtain large FWC even with narrow a MN, the MN potential should be kept larger for the entire MN. On the other hand, in order to suppress image lag, the electrical field for transfer from MN to FD should be kept larger. To meet both requirements, we successfully designed implant layouts and conditions for MN so that the potential has a continuous gentle slope from one edge to the other. Figure 5 shows a one-dimensional potential profile of the non-optimized MN and the well-designed MN in this work when TX2 is turned on.

### 3.3. Optical Design

The LP design is one of the most important factors to achieve high optical performances. As described Section 3.1, light sensitivity and its AR of the PD and the MN are a tradeoff relationship. For better light sensitivity of PD, light collection efficiency into the silicon should be maximized, while at the same time, for better PLS, the oblique light into the silicon should be minimized. Therefore, a new LP structure specialized for GS pixel is required.

We will discuss the desirable LP shape for a GS pixel referring to Figure 2. In order to collect oblique incident light, the upper surface of the LP (Lupper) should be maximized. It improves QE and AR. On the other hand, the bottom of the LP (Lbottom) must be smaller than the W aperture (Lbottom < Lw) in order to allow the light to enter into the W aperture without optical loss. When Lbottom is wider than the W opening, light is blocked by W. Then, quantum efficiency (QE) and AR decrease. Therefore, Lupper will be larger than Lbottom, and consequently, the LP has a tapered shape.

The design guideline for the tapered LP will be described with reference to Figure 6a. When the taper angle is α and the incident angle to LP taper is β, according to Snell’s law, if the first incident angle β is smaller than the critical angle, light will be refracted and leak out of the LP. Therefore, the taper angle α must be small as possible. Considering the multiple reflection, the incident angle to inside the LP decreases as the number of reflections increases. The second reflection angle is 2α smaller than the first reflection angle. Thus, the second reflected light will more easily leak out to the MN than the first reflected light. Figure 6b is a conceptual simulation image. If the LP is long and its taper angle is large, the leak light that enters the MN will increase.

From this study, the new design guidelines for the LP are as follows. First, the height of LP should be designed so that once light is totally reflected, it goes to the silicon surface through the bottom of LP. In other words, single total reflection in the LP is preferred. Second, the tapered angle of the LP should be designed to be small, considering oblique incident light. When F# is 2.8, since 10 degrees of light is incident, α needs to be less than 10.5 degrees.

In order to prove our design guidelines, we compared three types of LP shapes by 3D-FDTD (Finite Difference Time Domain) simulation, as shown in Figure 7.

(1)Type A: Long LP (α was 16 degrees).(2)Type B: Large taper angle (α was 20 degrees).(3)Type C: Small taper angle (α was 10 degrees).

At first, an incident angle of 0 degrees was simulated. The simulation was performed at a wavelength of 530 nm. We would like to emphasize that QE was comparable for all three structures. On the other hand, 1/PLS is greatly affected by the LP shape. Type A had the worst 1/PLS out of the three types. It indicates that a long LP is not appropriate for GS pixels. Type C gave the best 1/PLS.

Next, as shown in Figure 8, 1/PLS at the incident angle of 10 degrees was calculated. Simulated QE is almost equal for all three structures, even in oblique incident light. Type B, with the largest tapered angle, clearly showed the worst 1/PLS. From these results, type C was confirmed to be the best shape for the GS pixel. These simulated results indicate that our guideline is correct.

### 3.4. Experimental Result

The cross-section of the newly designed, fabricated 2.5 μm GS pixel with the small tapered LP is shown in Figure 9. The LP was successfully fabricated as designed. This 2.5 μm GS pixel used an N-type substrate in order to minimize the photodiode (PD) dark current.

The key pixel performances with color are summarized in Table 2. Linear FWC at PD was 6300 e^−^. FWC at MN was more than 9000 e^−^, which is enough to store electrons transferred from PD. Image lag was well suppressed. Temporal noise was 1.5 e^−^/s. The dark current and the standard deviation of the dark current at PD were 43 e^−^/s and 28 e^−^/s at 60 degrees, respectively. The dark current and the standard deviation of the dark current at MN were 13 e^−^/s and 24 e^−^/s at 60 degrees, respectively. The QE of a green pixel at 530 nm wavelength was 68%. The AR of the sensitivity was 12.5 degrees. 1/PLS of a green pixel at F# 2.8 tested with a halogen lamp was 10,400.

Details of the key performances discussed in Section 3.2 and Section 3.3 are described as follows. First, we discuss MN performances. Figure 10 shows the distribution of image lag at TX2 with the non-optimized MN and the well-designed MN. The MN in this work shows no outlier distribution of lag at TX2, which means that the image lag at TX2 was well suppressed.

We compared the dark current at the MN with that at the PD. Figure 11 shows the temperature dependence of the dark current. From this figure, the activation energy and doubling factor can be calculated, and the value was shown in the Table 3. The doubling factor of MN was comparable with that of PD, which means that the MN is fully pinned as well as the PD.

Next, the optical performances are described. Figure 12 shows the QE curves. The peak QE for a green pixel (530 nm) was 68%, and the peak QE of the monochrome was 78%. Figure 13 shows a normalized AR of QE in (a) the horizontal direction and (b) the diagonal direction, maintaining 80% of its peak value. In the horizontal direction, the AR achieved ±12.5 degrees. In the diagonal direction, although we adopted shared pixel architecture, which means asymmetrical pixel layouts, there is no difference between Gr and Gb pixels, because the incident light is successfully confined in the LP without reflection or absorption on metals or the poly-Si gates.

Figure 14 shows the AR of 1/PLS with a green pixel in the horizontal direction, which was maintained at half value or more in the range of ±10 degrees. Figure 15 shows the F# dependence of 1/PLS with a green pixel. 1/PLS was very stable down to F #2.8, and the value was 10,400.

Figure 16 shows an image capture by a 25 M pixel product with a 2.5 μm GS pixel array [15]. It was captured with an F# 0.95 lens and exposure time of 0.24 msec. There is no dynamic PLS artifact, even though the F# is 0.95.

Table 4 describes the comparison of the monochrome pixel with a previous report [14,16,17]. Despite the smaller pixel size compared with the previous report, the best in class performances were achieved.

## 4. Study of Near-Infrared Enhancement with Light Pipe Technology Using 2.8 μm and 3.2 μm GS Pixels

### 4.1. Effect of Light Pipe on MTF with Near-Infrared Enhancement

As discussed in previous sections, LP technology can confine incident light into a narrow region on the silicon surface. Reported approaches to enhance the quantum efficiency of NIR are (1) a thicker silicon photoresponse region [18], (2) adopting a reflection part [19], and (3) scattering structure [20]. For a GS pixel with a MN, approach (1) is preferred, because reflected or scattered light would go into the MN, which degrades 1/PLS significantly. A thicker silicon photoresponse region causes another issue of MTF, since angled incident light will cause electrons to be generated laterally far away from its impact point in the surface. LP technology can be applicable to realize good confined incident light into silicon and bring good MTF for NIR enhanced pixels. Figure 17 shows the simulation results of electric field distribution of incident light into 2.8 μm-pitch pixels with and without a LP when the wavelength of incident light is 850 nm. A smaller power of light close to the W-shield is clearly shown in Figure 17a compared with Figure 17b. That also brings confined light into the silicon body.

The above 2.8 μm pitch GS pixels were fabricated on 10 μm thick P-type wafers to evaluate MTF. As discussed in Section 2, PD consists of an N-type implanted region and a P-type epitaxial region. The P-type concentration of substrate in this experiment was the order of 10 to the 18^th^ power. Therefore, the lifetime of electrons in the substrate is short, and only an epitaxial region would contribute to NIR QE. In other words, an effective PD depth is 10 μm. The simulated one-dimensional potential profile of the center of the PD is shown in Figure 18. The potential is flat in the P-type epitaxial region where electrical cross-talk between adjacent pixels can happen.

Figure 19a shows a comparison of the measured MTF@50% contrast of a pixel with LP and without LP when the wavelength is 830 nm. These MTF values include the MTF of the camera lens. The MTF with LP was 1.6 times better than that without LP. Details of the spatial frequency dependence of MTF of a pixel with LP are shown in Figure 19b, where these MTF values do not include the MTF of the camera lens in order to compare with theoretical limits of the pixel. When the wavelength was 940 nm, the MTF@50% was 109l p/mm, and the QE at 940 nm of fabricated 2.8 μm GS pixel with LP was 11%. There is a difference between the measured MTF and the theoretical limit of MTF of 2.8-μm pitch pixel. That implies the existence of cross-talk, even with a LP, and the major factor would be electrical cross-talk due to the flat potential region.

### 4.2. Further MTF Improvement by Stacked Deep Photodiode Technology

For further improvement of MTF, electrical cross-talk should be reduced in addition to reducing optical cross-talk. In order to achieve that, a well isolated and deep PD is required. In order to realize such a PD, we already developed stacked deep photodiode (SDP) technology [21]. The simple cross-section schematic of the SDP structure is shown in Figure 20a. SDP consists of N-type epitaxial 1 and epitaxial 2 containing PDs and isolations. PD and isolation are formed in epitaxial 1, followed by the growth of epitaxial 2. Another PD and isolation are formed in epitaxial 2 with perfect alignment and connection to lower PD and isolation. In this report, we adopted the SDP technology to a 3.2 μm pitch GS pixel of which the PD depth was 7 μm. As shown in Figure 20b, the 7 μm PD is isolated from the N-type substrate. The isolation between PDs is also successfully formed. Therefore, the possibility of electrical cross-talk from an epi wafer deeper than PD and from adjacent pixels is well suppressed.

Figure 21 shows the spatial frequency dependence of the measured MTF at 940 nm wavelength not including the MTF of a camera lens. The measured MTF is consistent with the theoretical limit for a 3.2 μm pixel. Combining proper light confinement and reduced electrical cross-talk, good MTF performance is achievable, even in small pitch GS pixels.

Table 5 shows the key pixel performance of a 3.2 μm GS pixel with SDP technology. The NIR QE was successfully enhanced, and the QE value at 940 nm was 9%.

## 5. Conclusions

We developed the world’s smallest 2.5 μm GS pixel using a 65 nm process with an advanced LP structure. This GS pixel showed excellent optical performances: 68% QE at 530 nm, ±12.5 degrees AR, and 10,400 1/PLS with the F# 2.8 lens. In addition, we achieved an extremely low memory node dark current 13 e^−^/s at 60 °C by fully pinned MN.

Furthermore, we studied how the LP technology contributes to the improvement of the MTF in the NIR-enhanced GS pixel. A 2.8 μm GS pixel using a p-type substrate showed 109 lp/mm MTF/50% at 940 nm, which is 1.6 times better than without LP. The MTF can be more enhanced by the combination of the LP and the successfully isolated PD. We showed results from the 3.2 μm GS pixel with SDP technology showing 156 lp/mm MTF at 940 nm, which is the theoretical value.

## Figures and Tables

**Figure 1 sensors-20-00307-f001:**
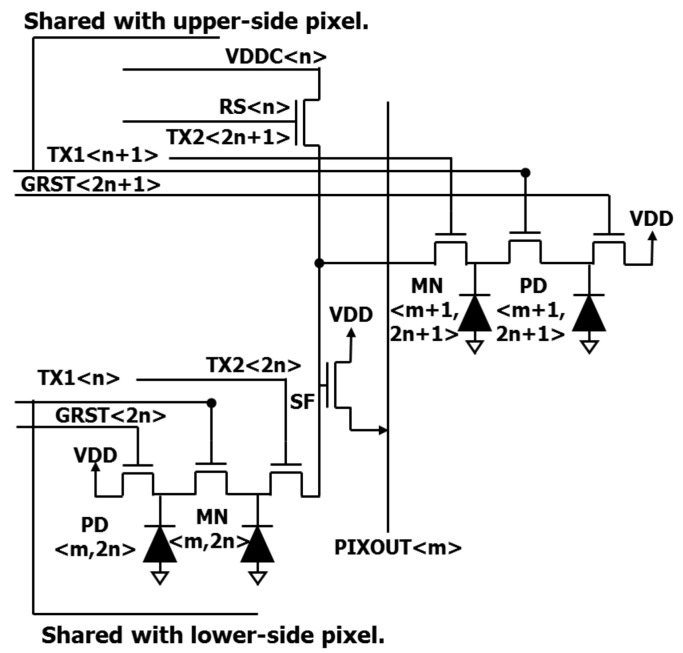
Pixel circuit schematic of global shutter (GS) pixel.

**Figure 2 sensors-20-00307-f002:**
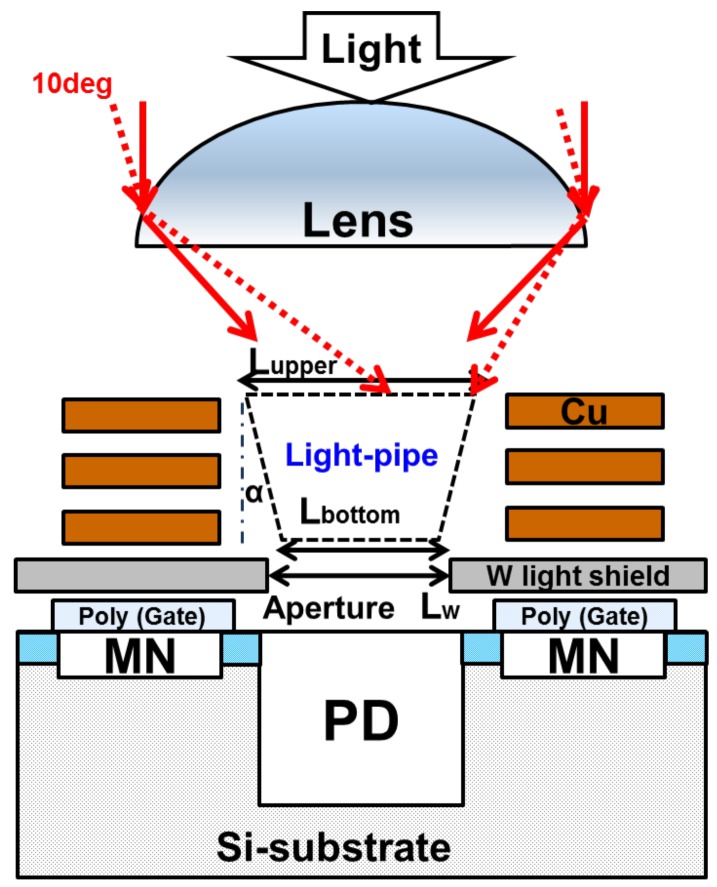
Cross-section schematic of the global shutter (GS) pixel.

**Figure 3 sensors-20-00307-f003:**
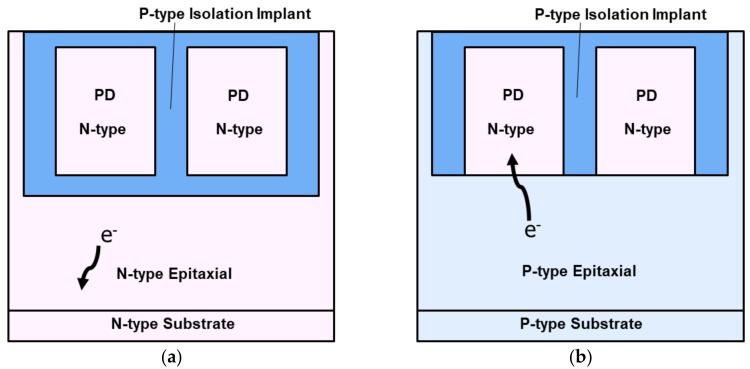
Simple photodiode (PD) schematic with (**a**) N-type substrate, (**b**) P-type substrate.

**Figure 4 sensors-20-00307-f004:**
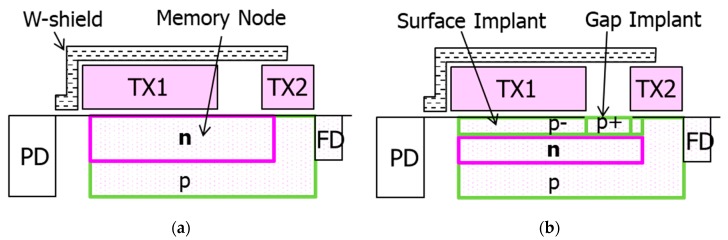
Cross-section schematics of (**a**) conventional memory node (MN) and **(b**) full pinned MN.

**Figure 5 sensors-20-00307-f005:**
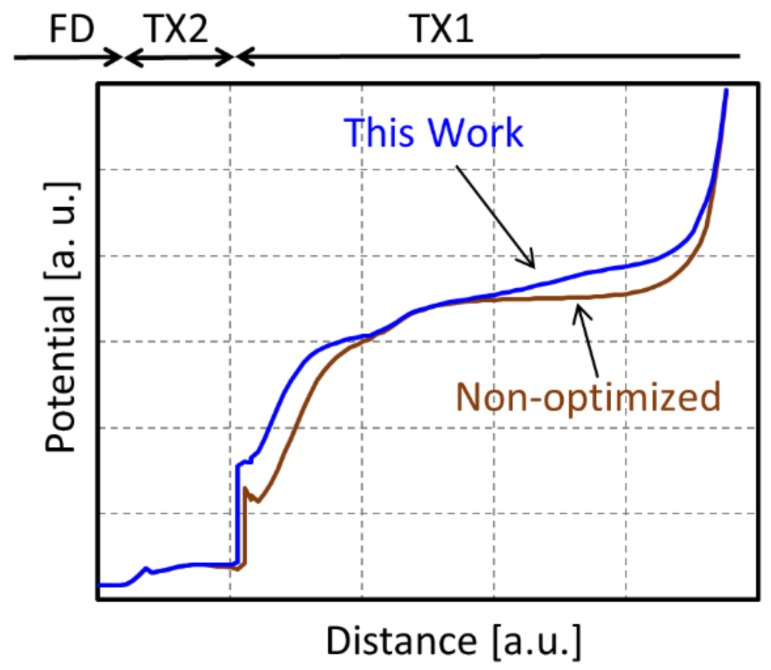
One-dimensional potential profile of the non-optimized MN and the well-designed MN.

**Figure 6 sensors-20-00307-f006:**
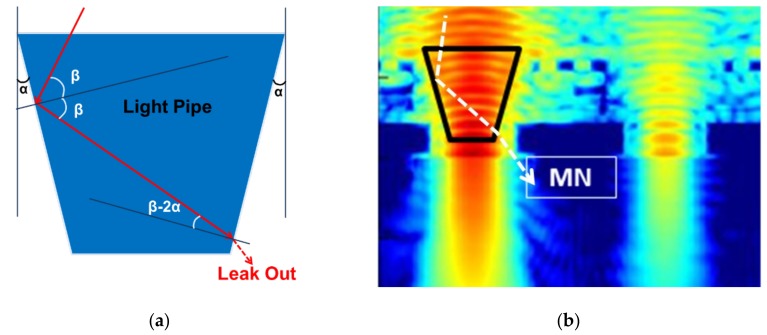
Schematic of reflection: (**a**) explanation of incident angle and (**b**) conceptual simulation image.

**Figure 7 sensors-20-00307-f007:**
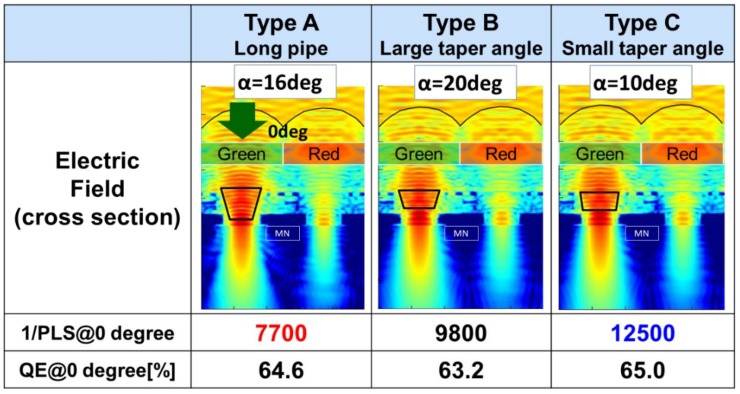
Comparison of three types of light pipes at an incident angle of 10 degrees.

**Figure 8 sensors-20-00307-f008:**
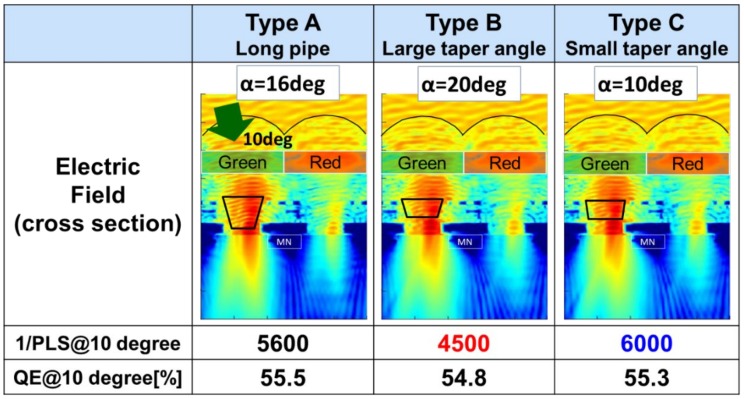
Comparison of three types of light pipes at an incident angle of 10 degrees.

**Figure 9 sensors-20-00307-f009:**
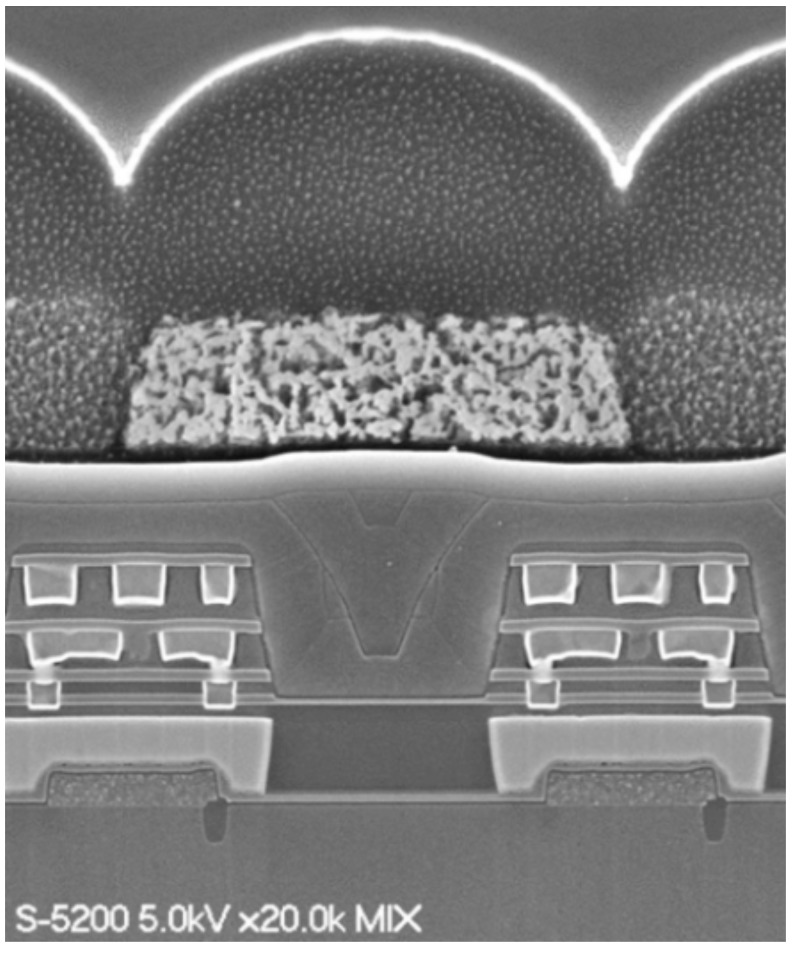
Cross-section of 2.5 μm GS pixel.

**Figure 10 sensors-20-00307-f010:**
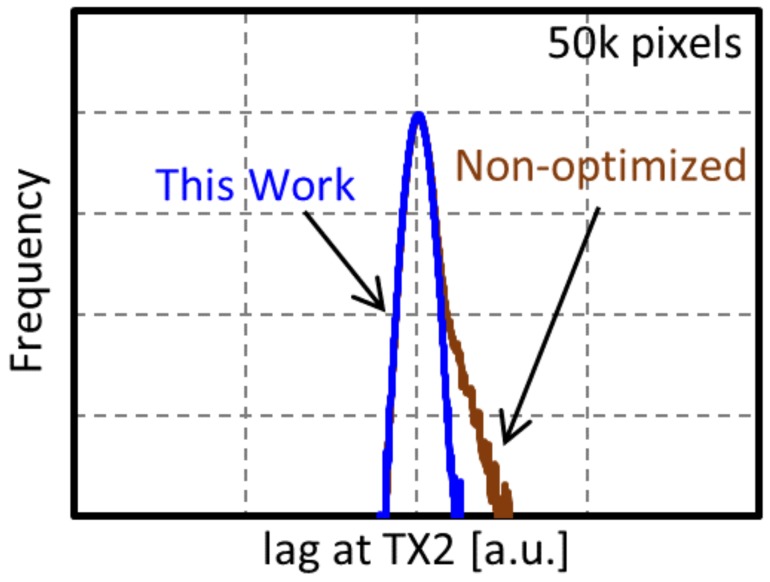
Image lag distribution at TX2.

**Figure 11 sensors-20-00307-f011:**
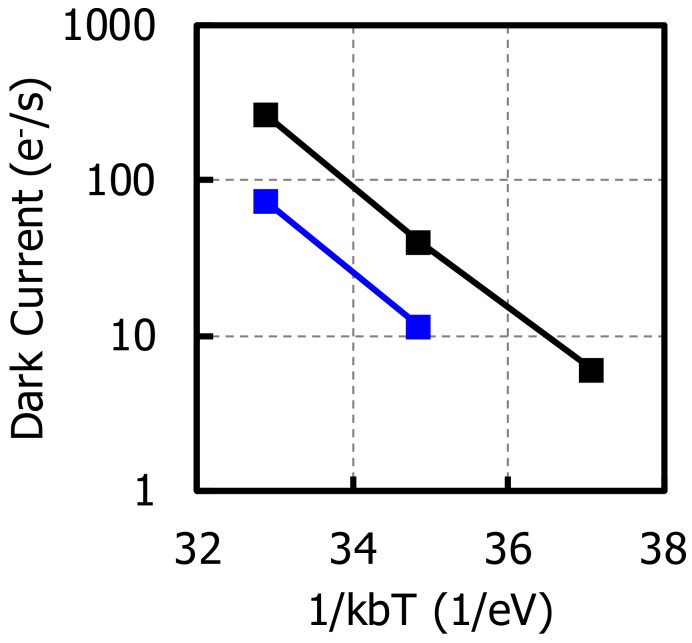
Temperature dependence of the dark current.

**Figure 12 sensors-20-00307-f012:**
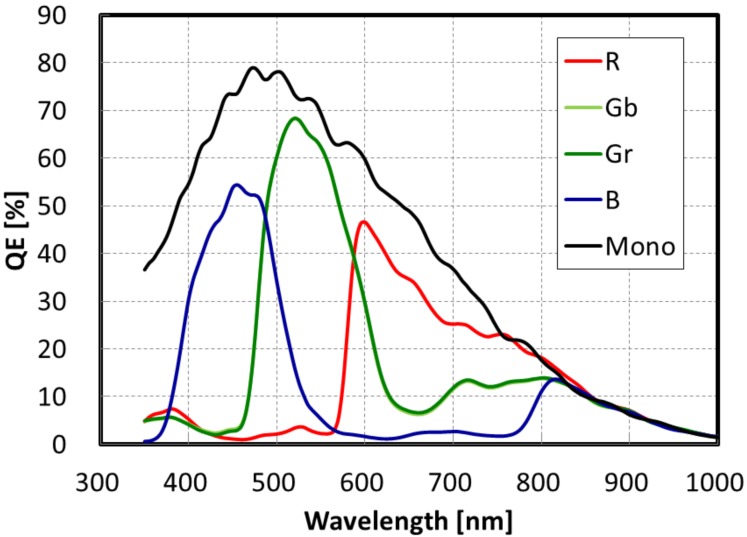
Quantum efficiency (QE) curves of the 2.5 μm GS pixel.

**Figure 13 sensors-20-00307-f013:**
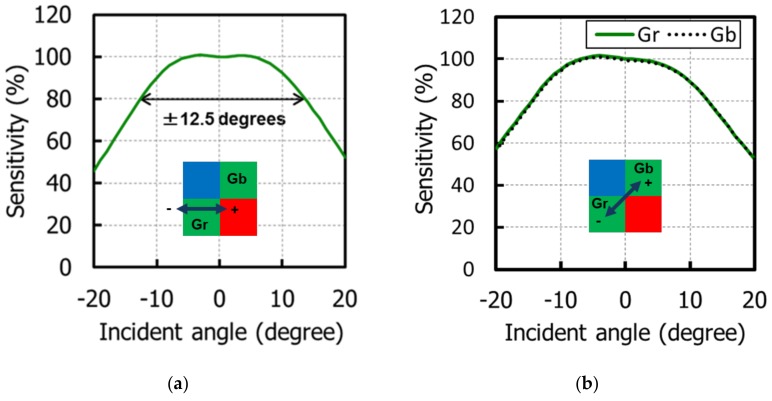
Normalized angular response of QE in (**a**) the horizontal direction and (**b**) the diagonal direction.

**Figure 14 sensors-20-00307-f014:**
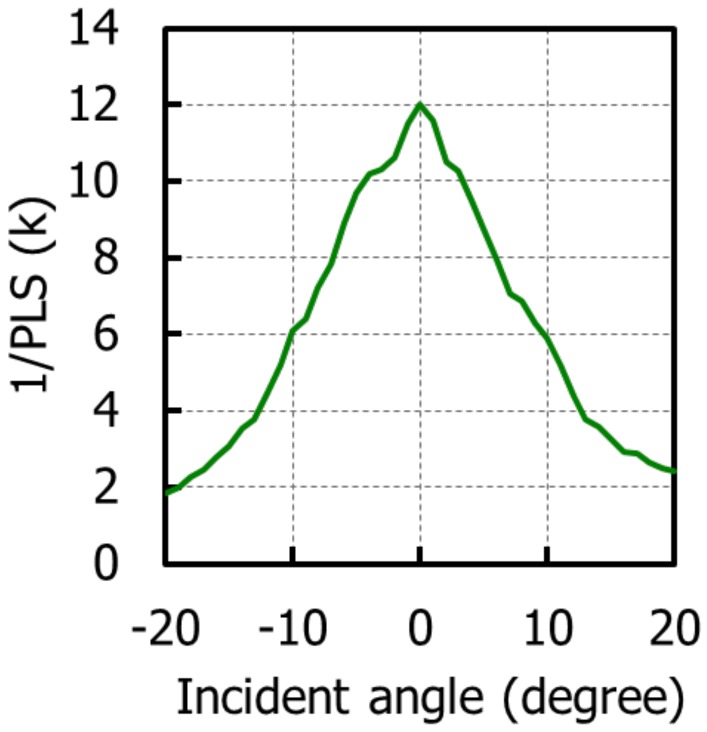
Angular response of 1/PLS (parasitic light sensitivity) with a green pixel in the horizontal direction.

**Figure 15 sensors-20-00307-f015:**
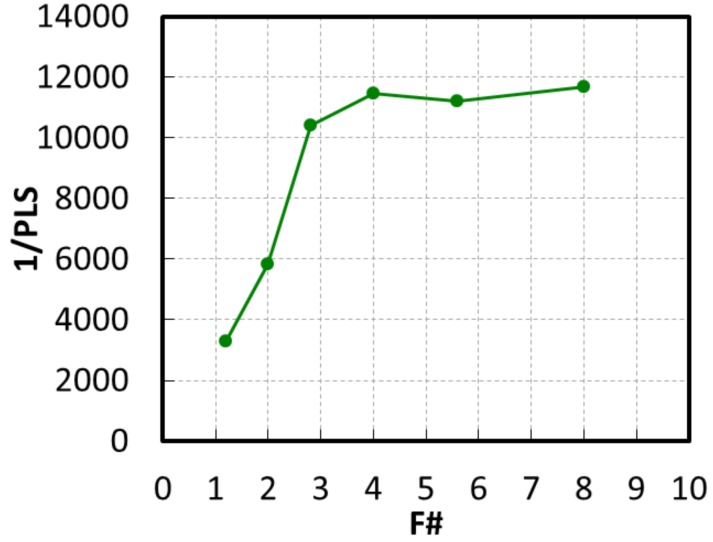
F# dependence of 1/PLS with a green pixel.

**Figure 16 sensors-20-00307-f016:**
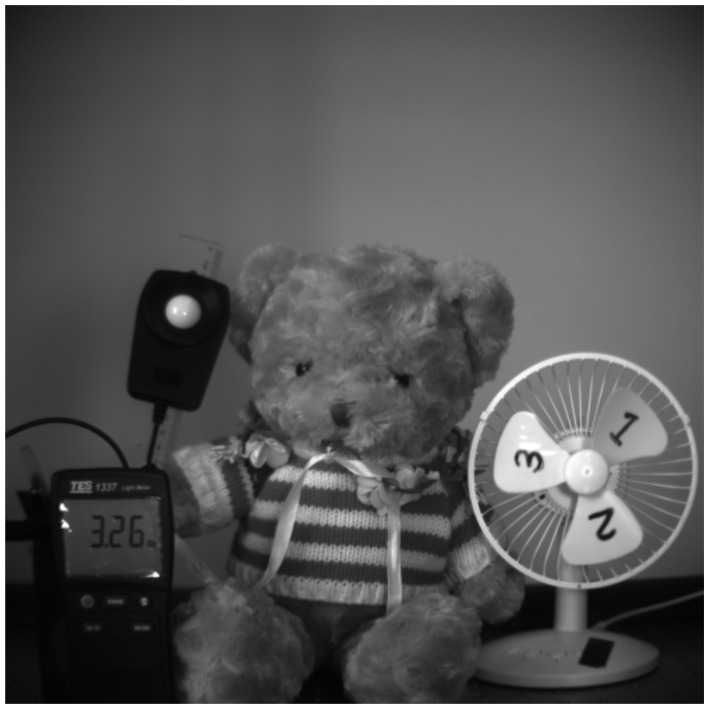
Snapshot of 2.5 μm GS pixel captured with an F# 0.95 lens and exposure time of 0.24 msec.

**Figure 17 sensors-20-00307-f017:**
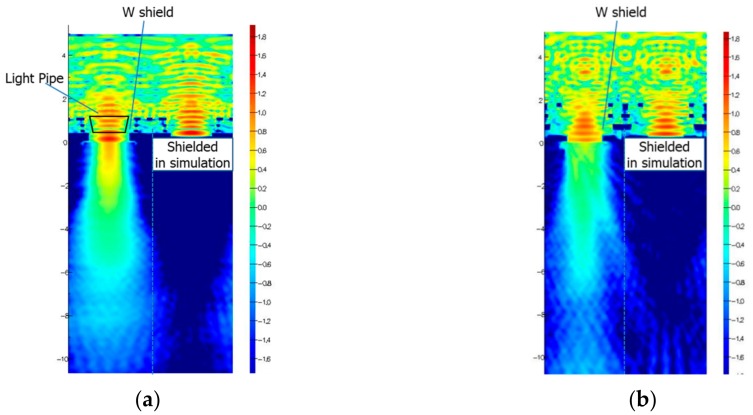
Comparison of electric field of incident light at 850 nm (**a**) with light pipe and (**b**) without light pipe. The right-side pixel is shielded by metal on purpose to extract light intensity only from the left-side pixel into the substrate.

**Figure 18 sensors-20-00307-f018:**
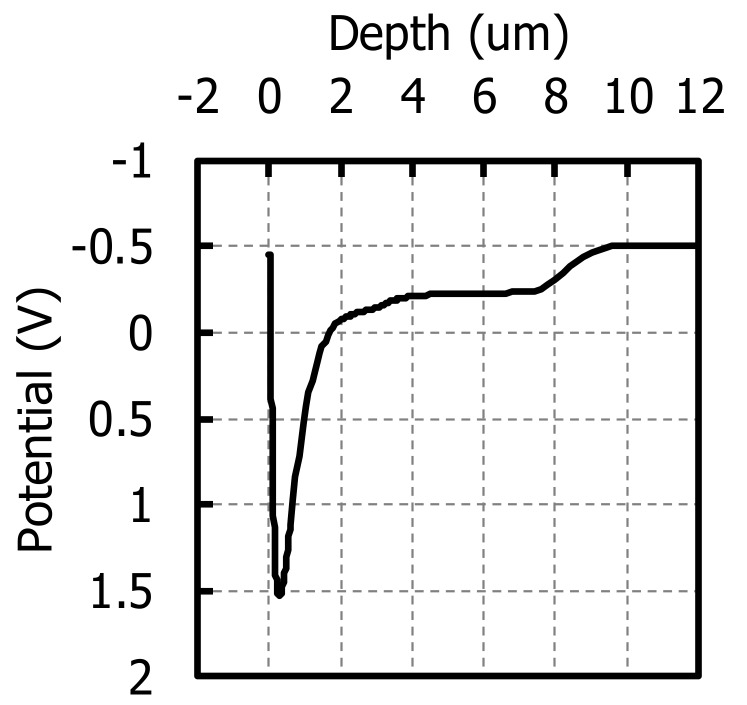
One-dimensional potential profile of the center of the PD.

**Figure 19 sensors-20-00307-f019:**
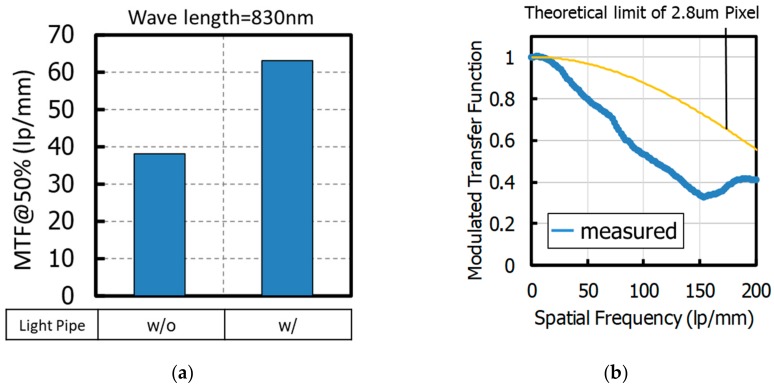
(**a**) Comparison of an MTF@50% contrast of a pixel with a light pipe and without a light pipe (wavelength is 830 nm, including the MTF of the camera lens). (**b**) Spatial frequency dependence of the MTF of a pixel with a light pipe (wavelength is 940 nm not including the MTF of a camera lens). MTF: modulation transfer function.

**Figure 20 sensors-20-00307-f020:**
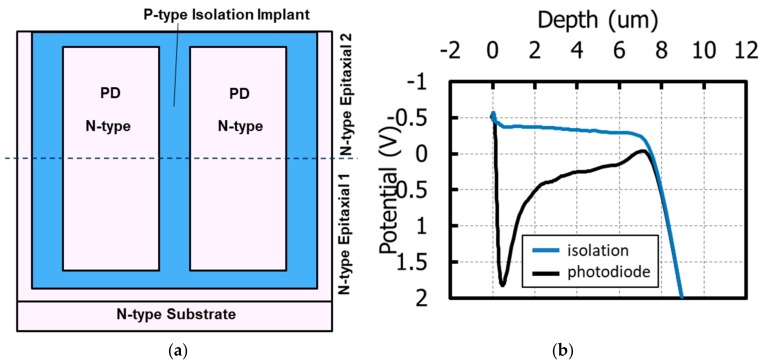
(**a**) Schematic of the cross-sectional structure of a stacked deep photodiode, (**b**) one-dimensional potential of a stacked deep photodiode.

**Figure 21 sensors-20-00307-f021:**
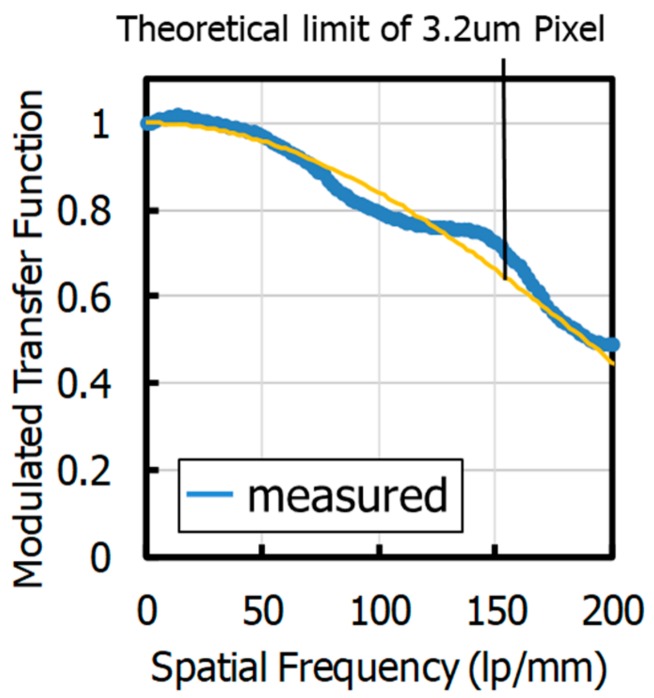
Spatial frequency dependency of MTF of a 3.2 μm pitch GS pixel with a light pipe and stacked deep photodiode.

**Table 1 sensors-20-00307-t001:** Figure of merit for charge domain.

	Photo Diode (PD)	Memory Node (MN)
Light Sensitivity	Lager-is-better (QE)	Smaller-is-better (PLS)
Angular Response of Sensitivity	Lager-is-better	Smaller-is-better
Full Well Capacity	Lager-is-better	Larger than PD
DC, DSNU	Smaller-is-better	
Lag	Smaller-is-better	

**Table 2 sensors-20-00307-t002:** Key pixel performances with color.

Pixel Performance		Unit	2.5um
Conversion Gain @SF out		uV/e^−^	100
Linear FWC @PD (max SNR)		e^−^	6300
Saturated FWC @ MN		e^−^	>9000
Image Lag @ TX2		e^−^	0
Image Lag @ TX1		e^−^	0
Image Lag @ GRST		e^−^	0
Temporal Noise		e^−^ rms	1.5
Dark Current	@ PD (60C)	e^−^/s	43
Standard Deviation		e^−^/s	28
Dark Current	@ MN (60C)	e^−^/s	13
Standard Deviation		e^−^/s	24
QE @ Green (λ = 530 nm)		%	68
AR of Sensitivity		degree	12.5
1/PLS @ F2.8		-	10,400

**Table 3 sensors-20-00307-t003:** Activation energy and doubling factor of the dark current.

	Ea (eV)	Doubling Factor (°C)
MN	0.98	5.5
PD	0.95	5.8

**Table 4 sensors-20-00307-t004:** Comparison table with previous report. FWC: full well capacity, QE: quantum efficiency.

	Unit	This Work	[14] IISW2013	[16] IISW2017	[17] IEDM2018
Pixel Pitch	μm	2.5	2.8	3.4	3.2
Linear FWC (Per pixel area)	e^−^ (e^−^/μm^2^)	6300 (1008)	6000 (765)	6100 (528)	7100 (693)
Dark Current at MN	e^−^/s	13	60	-	5
Peak QE (mono)	%	78	70	62	72.9
Sensitivity	e^−^/lx s	30,800	-	28,000	-
1/PLS (mono)	-	8100	2200	28,000	3333 @λ = 505 nm

**Table 5 sensors-20-00307-t005:** Key pixel performances of 3.2 μm GS pixel with stacked deep photodiode.

Pixel Performance		Unit	3.2 um
Conversion Gain @SF out		uV/e^−^	80
Linear FWC @PD (max SNR)		e^−^	10,200
Saturated FWC @ MN		e^−^	>14,000
Image Lag @ TX2		e^−^	0
Image Lag @ TX1		e^−^	0
Image Lag @ GRST		e^−^	0
Temporal Noise		e^−^ rms	1.6
Dark Current	@ PD (60C)	e^−^/s	66
Standard deviation		e^−^/s	54
Dark Current	@ MN (60C)	e^−^/s	25
Standard deviation		e^−^/s	12
QE @ λ = 940 nm		%	9
AR of Sensitivity		degree	16
MTF @50%, λ = 940 nm		lp/mm	156 (theoretical limit)

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
