# Peer review of "A High-Performance 2.5 μm Charge Domain Global Shutter Pixel and Near Infrared Enhancement with Light Pipe Technology [Author-notes fn1-sensors-20-00307]"

_sensors, 2020, doi:10.3390/s20010307_

Round 1

Reviewer 1 Report

File attached

Author Response

Dear Reviewer,

I greatly appreciate your kindly review.

I revised my manuscript according to your comments.

Please see the attachment and revised manuscript.

Kind regards,

Mizuno

Reviewer 2 Report

It is a nice manuscript with content that would be of interest to imaging sensor community. The manuscript requires though some minor modifications, fixing typos and some logical constructs to make it more clear for the audience. I will try to highlight what noticed below.

Throughout of the text please use “ µm” instead of “um”. There is space between number and units, ex. 2.5 µ Page 1, line 21: “plat form” => “platform”. Page 1, lines 23 and 24: First use of QE, AR, PLS abbreviations. Please provide full text, ex. quantum efficiency (QE), etc. Page 1, line 24: “extremely low dark current” => “extremely low pixel memory dark current”. To distinguish it from photodiode dark current. Page 1, line 29: “the successfully isolated photodiode”. More clarification in logical construct may be needed to highlight “successfully”. Because PD isolation should be successful for pixels to function properly. May be something about improved optical and electrical PD isolation? Page 1, line 31: “improvement of more than 100%”. Improvement of what? Sensitivity? Page 2, line 53: “these applications”. Not clear which “these” applications. Page 2, line 55: “cross talk to adjacent pixels – namely modulated transfer function (MTF)”. “Crosstalk”. Generally MTF is not crosstalk but a measure that reflects pixel crosstalk. Better sentence construct is needed to make it clear. Page 2, line 58: “parasitic light sensitivity (PLS)” already was abbreviated above. Page 2, line 58-59: “Its small metal window… to have good MTF due to diffraction at W-shield”. This is debatable. You may be surprised to learn that MTF dependency on the W-shield is not that large. Metal reflections may cause more photons to enter neighboring pixels but may be not large quantities. Definitely not due to diffraction effects. Lightguide helps to reduce these reflections by funneling more photons within the pixel to PD. Page 2, line 67: “narrow MN design” => “narrow shaped MN design”. Page 2, line 76: “from MN to PD” => “from MN to FD”. Page 3, line 80: “[11]”. Wrong reference? [11] does not have either the pixel architecture nor cross section of the pixel. Page 3, line 92: “electrically isolated from the …” => “better electrically isolated from the …”. P-type substrate and p-implants also electrically isolate PDs, etc. Page 3, line 96-97: “a much higher NIR QE than that with high P-type concentration of substrate instead of higher dark current”. Need better logical construct as not clear relation between NIR sensitivity and dark current. Generally, NIR sensitivity is driven mainly by photon collection depth (related to epi thickness) and to less degree to dopant concentration. Page 4, Table 1: Angular Response of Sensitivity for PD “Larger-is-better” and for MN “Smaller-is-better”? Page 4, line 112: “(2) LP design” =>”(2) LP and metal shield design”. Both LP and W-shield play hand in hand for better GSE. Page 4, line 116, 120, 121: Replace “under the gap” to “in the gap”. Page 7, line 184, 185: “DSNU… 28 e-/s” and similar about DSNU. Generally DSNU is not measured in e-/s as it always have some floor and is dependent on both temperature and integration time. Usually it is expressed as electrons at specific temperature and integration time. Also numbers shown are too small. For example, my interpretation for DSNU 28 e-/s at 16.66ms integration time would be 28/60 ~ 0.45 e- which is too small at 60C to be trustful. Please clarify and change to proper units and numbers. Table 2: DSNU units and numbers need adjustment and clarification. See above. Page 8, line 193: “lag at TX2 was well suppressed”. Actually there is indication of some small single electron average level lag present in the pixel distribution histogram. Would be nice to have actual lag measured numbers in electrons in Table 2 and Table 5. Page 8, line 196: “activate energy” => “activation energy”. Page 9, line 202: “Activate energy” => “Activation energy”. Page 9, line 204: “The QE for Green” => “The peak QE for Green”. Page 9, line 205: “Figure 13 shows AR” => “Figure 13 shows normalized AR”. Page 10, line 213: “Figure 13. Angular response” => "Figure 13. Normalized angular response”. Page 10, line 215, 217, 218: “1/PLS of the color” or “1/PLS with color”. Not clear what color? Green, green pixel? Page 11, line 219-220: This whole sentence need more clear construct. Consider rephrasing or dividing into two. This is not “snapshot of 2.5 µm pixel” but rather image capture by a 25Mp 2.5 µm pixel array. Page 11, Table 4: “Dark Currentat” => “Dark Current”. Page 12, line 240: “shielded by metal intently” => “shielded by metal on purpose”. Page 12, line 253: “MTF of camera lens to in” => “MTF of camera lens in”. Page 12, line 255: “There is discrepancy” => “There is difference”. Page 13, line 260: “MTF of camera lens.” => “MTF of camera lens).” Page 13, line 274: “of PD with stacked deep photodiode” => “of stacked deep photodiode”. Page 14, line 277: “good MTF property” => “good MTF performance”. On Figure 21 would be nice to discuss why measured MTF in some places is better than theoretical MTF, ex. show potential impact of the MTF measurement procedure and/or setup on accuracy. Page 14, line 282: “and the value at 940 nm” => “and the QE value at 940 nm”. Table 5 need clarification and checking on DSNU units and numbers. Page 14, line 288: “low dark current 13 e-/s” => “low pixel memory dark current 13 e-/s”. Page 15, line 291 and 294: “lp/mm MTF at 940nm” => “lp/mm MTF/50% at 940 nm”. Page 15, line 292: “better than that without” => “better than without”. Page 15, line 336-337: When, where, pages for the reference? Page 16, line 343-345: Not clear where it was published, organization, pages, etc.

Author Response

Dear Reviewer,

I greatly appreciate your review.

I revised my manuscript according to your comments.

Please see the attachment and revised manuscript.

Kind regards,

Mizuno

Reviewer 3 Report

This manuscript described the start-of-the-art on the charge mode global-shutter pixel development.

To implement the charge mode global shutter pixel in BSI process has the difficulties on isolate the parasitic light leakage to the charge storage node. Thus, charge mode global shutter pixel in FSI process is a better approach. However, it will has lower sensitivity. 

This pixel design and fabrication has reasonable sensitivity and peak QE, however, the parasitic light rejection ratio is still in the low side. It will not meet the requirement of image operating in certain high illumination conditions. 

Author Response

Dear Reviewer,

I greatly appreciate your kind review.

Kind regards,

Mizuno